# Genome-Wide, High-Density Genotyping Approaches for Plant Germplasm Characterisation (Methods and Applications)

**DOI:** 10.3390/ijms262411833

**Published:** 2025-12-08

**Authors:** Sirine Werghi, Brian Wakimwayi Koboyi, David Chan-Rodriguez, Hanna Bolibok-Brągoszewska

**Affiliations:** Department of Plant Genetics Breeding and Biotechnology, Institute of Biology, Warsaw University of Life Sciences, 02-776 Warsaw, Poland; sirine_werghi@sggw.edu.pl (S.W.); brian_koboyi@sggw.edu.pl (B.W.K.); david_chan_rodriguez@sggw.edu.pl (D.C.-R.)

**Keywords:** genotyping, plant genetic resources, genetic diversity, GBS, reduced representation sequencing, whole genome resequencing, DNA-genotyping arrays

## Abstract

Germplasm collections are a treasure trove of humanity. The accessions constituting those collections (wild crop relatives, landraces, cultivars, etc.) contain genes and allelic variants, which evolved prior to or post domestication, in the course of adaptation and selection, and can be used in breeding to address current and future needs. Precise characterisation of genetic diversity is essential for the efficient conservation of genetic resources and their effective utilisation in crop improvement. Detailed genetic profiles resulting from DNA genotyping constitute a basis for establishing the level of genetic diversity of a collection, analysing population structure, identifying redundancies, performing genome-wide association scans (given the availability of phenotypic information), detecting loci under selection, and many other applications. To obtain an accurate picture of genetic diversity (at the DNA sequence level), robust, high-density, high-throughput, and cost-effective methods are needed. With the advances in the next-generation sequencing, new genotyping approaches emerged (such as genotyping-by-sequencing, whole genome resequencing), which provide excellent genome coverage and low cost per datapoint (with tens of thousands to millions of loci analysed in a single assay). Crop-specific, custom, microarray-based genotyping solutions were also developed. The aim of this review is to provide a comparative description of the genome-wide, high-density genotyping technologies that are most frequently used nowadays, comprising their advantages and drawbacks, as well as factors that determine, which of the methods will best suit the particular germplasm characterisation project. Further, we characterise the current role of these methods in addressing the challenges related to the effective management and use of genetic resources and present recent examples of their application in selected crop plant groups. Finally, we briefly describe constraints to germplasm characterisation and future prospects.

## 1. Introduction

Over 5.9 million plant accessions (predominantly crop plants and their wild relatives) are maintained in ex situ germplasm collections worldwide, according to the III Report on the State of the World’s Plant Genetic Resources for Food and Agriculture [1]. Together these accessions constitute a living library of natural genetic variation—gene variants and specific allelic combinations that came into existence over the course of the evolution of plant species, adaptation to specific geoclimatic conditions, in response to natural or artificial selection pressure.

Crucially, plant genetic resources (PGR) contain gene variants, which could be useful for crop improvement, to address current needs related to, among others, to climate change (crop varieties more resilient to hostile conditions, such as drought, heat are needed), and population growth (higher yields have to be achieved to provide enough food for the constantly growing human population, for example, by introduction of genes increasing nutrient use or/and uptake efficiency). A well-known and striking example of the realisation of the potential of PGR are the Green Revolution genes [2] originating from landraces of wheat and rice. Semi-dwarf, high-yielding varieties carrying these genes significantly contributed to improving food security in the developing countries between 1960 and 2000 [3]. More recent examples of PGR potential in crop improvement include the discovery of *S. pennellii* genome segments that enhanced the productivity of a top market variety by more than 50% [4]. In rice, a gene enhancing phosphorus deficiency tolerance, *Pstol1*, was discovered in a traditional variety, Kasalath. This gene is absent from the reference genome of the variety Nipponbare and other modern phosphorus-deficiency-sensitive rice varieties [5].

Unfortunately, there are considerable constraints to the effective use of PGR in plant improvement. The vast size of PGR collections often means that little is known about the extent and the distribution of the diversity, and it is very challenging to pinpoint an accession or accessions that could contribute variation suited to the particular breeding target. Specific challenges resulting from the biology of the species in question (such as outcrossing nature, high heterozygosity, heterogeneity of accessions), complicate the issue further [6,7]. Specific schemes (such as development of NAM, MAGIC, or RCSL populations [8,9] were devised to facilitate inclusion of useful exotic variation into breeding programmes. Nevertheless, the accurate choice of the PGR accession for each of these approaches is fundamental to the success of the endeavour and can be facilitated by the availability of the genome-wide, high-density genotypic information for the PGR collection in question. 

The aim of this review is to provide a comparative description of the genome-wide, high-density genotyping technologies that are most frequently used nowadays, summarising their advantages and drawbacks, as well as discussing factors that determine, which of the methods will best suit the particular germplasm characterisation project. Further, we characterise the current role of these methods in addressing the challenges related to the effective management and use of genetic resources and present recent examples of genome-wide high-density genotyping application in selected crop plant groups. Finally, we briefly describe constraints to germplasm characterisation and future prospects.

## 2. Plant Germplasm Genotyping Approaches

Various methods were developed over the decades to detect polymorphism at the DNA level, starting from restriction enzyme-based methods (RFLP), various PCR-based methods (RAPD, SSR), to methods combining both digestion and PCR (such as AFLP). In the first decade of the XXI century, microarray-based methods, especially those independent from the availability of sequence information for the species in question, such as DArT [10] became very popular. From the above-mentioned methods, SSRs still enjoy considerable popularity [11,12,13] as a reliable and affordable tool for detecting DNA polymorphism, particularly when funding is limited. SSRs have the advantages of being co-dominant, highly polymorphic and multiallelic, as well as sequence specific and therefore easily transferable between experiments. On the other hand, the development of new SSR markers is usually a laborious task and the throughput is low, since typically a single SSR locus is targeted by a single assay [14]. Though the number of loci that can be analysed in a single study is not very high (usually around 15–30), due to a relatively high labour-intensiveness of the assay, a number of carefully chosen SSR markers can provide reliable information on the genetic diversity level and its distribution in a given collection [15,16,17,18]. SSRs also remain an important tool in specific applications, e.g., in varietal identification. In apple a Malus UNiQue genotype coding system (MUNQ) based on 15 SSR [19] is used for germplasm identification.

The advances in next-generation sequencing technologies gave rise to a new type of genotyping methods, described as reduced-representation sequencing methods [20,21,22,23,24]. With steady improvements in sequencing capacity, sinking prices, and ongoing progress in bioinformatic analysis tools and pipelines, detection of polymorphism via low-coverage whole genome resequencing (WGRS) became feasible [25,26,27,28,29]. The constantly growing availability of sequence data and the discovery of numerous single nucleotide polymorphisms (SNPs) permitted the development of microarray-based methods aimed at simultaneous detection of polymorphism at thousands of preidentified, selected SNPs such as Illumina Infinium Beadchip, Affymetrix Axiom or Affymetrix GeneChip [30]. These three groups of genotyping methods prevail nowadays and will be characterised in the Section 2.1 of this review. An overview of these methods is presented in Table 1.

Specific methods of genotyping a moderately sized set of selected SNPs (for instance, Kompetitive Allele-Specific PCR (KASP) [31] and Fluidigm genotyping assays [32]) or analysing sequence diversity of selected target genes (for example, by amplicon sequencing [33] are also in frequent use, but since these methods do not provide high-density genome-wide coverage, they will not be described here).

### 2.1. Reduced Representation Sequencing (RRS) Methods

Reduced representation sequencing (RRS) methods, such as GBS [20], DArTseq [21], RAD-seq [23], SLAF-seq [34], sometimes also collectively called GBS [24] provide high-density, low-cost genotypic data. Since a reference genome sequence is not required, these methods are particularly useful for non-model plants and orphan crops. Generally, these methods permit simultaneous discovery and genotyping of variants.

RRS workflow includes several key steps: (1) genome complexity reduction using restriction enzymes; (2) barcoding genomic DNA with indexed adaptors; (3) high-throughput sequencing of barcoded fragments; (4) bioinformatics analysis to identify genetic variants. Genome complexity reduction is a central concept of the reduced representation sequencing methods. In this analysis step, also called genome sampling, a subsection of the genome, usually containing gene-rich regions, is selected. This is achieved in most cases by using a methylation-sensitive restriction endonuclease for DNA digestion, which initiates the sample preparation. Then, ligation of adaptors, size selection, and PCR amplifications (depending on the approach) can be used to achieve the final set of fragments representing a given DNA sample, which will be subjected to NGS [22]. Bioinformatic pipelines are available for analysis of resulting reads and include both reference-based and de novo solutions, when a reference genome sequence is not yet available for the species being analysed [35].

Reduced-representation approaches typically begin with raw read demultiplexing and adapter/quality filtering, followed by alignment to a reference genome (or pseudo-reference) and variant calling. Several software packages are commonly used for GBS and related reduced-representation data analysis. The TASSEL-GBS pipeline [36] was originally created for maize and allows efficient SNP discovery and genotype calling in large sample sets with low computational requirements. Within TASSEL 3.0, the UNEAK module [37] provides a reference-free option for species lacking a genome assembly. However, the UNEAK software module assumes diploidy, which can limit its performance in polyploids. Another popular toolkit, Stacks (http://catchenlab.life.illinois.edu/stacks/manual/, 2 December 2025) [38], has both de novo and reference-based workflows for SNP discovery. It has been applied across many plant species, especially when high-quality reference genomes are not available. For variant calling in complex or polyploid species, FreeBayes (https://github.com/freebayes/freebayes, access date: 2 December 2025) [39] provides a Bayesian framework. This allows users to specify ploidy level and model allele dosage more accurately. For deeper or higher-coverage datasets, generic variant-calling tools like GATK (https://gatk.broadinstitute.org, access date: 2 December 2025) [40] are also frequently used.

Most imputation algorithms were initially created for the human diploid genome and then adapted to plant genome analysis. In these genomes, each locus has two alleles, and researchers can infer haplotype phase from linkage disequilibrium patterns. The newest popular software successor is Beagle 5.0 [41]. Beagle performs haplotype-based imputation efficiently in large datasets and has been used with crops like soybean [42]. Other tools, such as IMPUTE5 [43] or its versions [44], also depend on pre-phased reference panels. They are best suited for diploid or highly inbred populations where high-quality reference genotypes are available [44]. The module FILLIN created for plant breeding within the TASSEL 5 framework [36] offers a method based on the haplotype library. It efficiently imputes missing genotypes in diploid datasets but assumes biallelic loci. These tools perform well in diploids because estimating allele dosage and phasing is relatively simple compared to polyploids.

In polyploid species, imputation is more difficult because multiple homologous chromosome sets make it harder to estimate allele dosage, phase haplotypes, and model segregation [45]. To address this, several tools have been created specifically for or adapted to polyploids, such as Imputef [45] and polyRAD (https://github.com/lvclark/polyRAD, access date: 2 December 2025) [46]. In TASSEL5, FILLIN [47] can be used with higher ploidy levels through haplotype libraries, although accuracy may decrease when allele dosage is uncertain. Ongoing development of polyploid-aware imputation and variant-calling tools is still limited because realistic models of complex genomes (featuring multiple chromosome copies, allele dosage variation, homoeologous sequence similarity, and sometimes mixed inheritance patterns) require high computing power [48].

Quality control is mostly performed using VCFtools (https://github.com/vcftools/vcftools, access date: 2 December 2025) [49] or PLINK (https://www.cog-genomics.org/plink, access date: 2 December 2025) [50], filtering markers by call rate, minor allele frequency, and Hardy–Weinberg equilibrium before conducting downstream analyses such as GWAS, diversity, or population-structure inference.

Depending on the species, GBS delivers thousands to hundreds of thousands of SNPs per sample [51]. In a recent study on faba bean, the GBS cost per sample (including library preparation and sequencing has been reported at approximately EUR 21 per sample [52], which translates to a very low cost per data point.

Since GBS methods do not rely on a set of pre-selected SNPs, they are consequently less prone to an ascertainment bias, a problem associated mostly with SNP arrays, which will be described below. However, GBS may introduce biases related to the choice of the restriction enzyme [22,52]. Wickland et al. [53], showed that SNP call sets from various GBS pipelines differed considerably, highlighting that comparability across studies is not guaranteed unless the method is exactly the same. Moreover, the reproducibility of GBS is dependent on the experimental protocol: Zamalutdinov et al. compared different restriction enzyme combinations (*HindIII*-*NlaIII*, *PstI*-*MspI*, and *ApeKI*) and different SNPs calling pipelines in 12 soybean varieties. They found that the enzyme selection and the choice of the right pipeline inherently influence the number of SNPs and their quality [54].

Although RRS methods enable deep sequencing of small genome portions in many samples, certain genomic regions, which might be critical for reaching very specific study aims (such as loci targeted by selection or associated with adaptive divergence) may be missed by the given assay design [55]. Further drawbacks of these methods include uneven coverage, higher rates of missing data, and under-calling of heterozygotes, especially in heterozygous or polyploid species. These issues can be mitigated through the use of genotype imputation and improved bioinformatic pipelines [53].

Noteworthy, it was demonstrated recently that the capacity of the polymorphism detection based on short sequence reads can be extended beyond primarily SNPs with the help of long-read sequencing data [56]. In this study, low coverage (~12×) Oxford Nanopore data from a moderately sized set of soybean accessions (17) was used for SV discovery and those variants could then be genotyped with high accuracy using Illumina reads (WGRS) in a population consisting of 102 Canadian soybean cultivars.

Despite certain limitations, RRS is an immensely popular genotyping approach, widely used for germplasm characterisation and genomic studies in a broad range of plant species. For many applications, such as phylogeny and population structure analysis, GWAS, and genomic prediction, the density and quality of GBS data are sufficient to obtain very reliable results [57] comparable with the outcomes of WGR [58], but in a far more cost-effective way. Examples of plant germplasm characterisation studies involving RRS are listed in Table 2.

### 2.2. Whole Genome Resequencing (WGRS)

WGRS involves sequencing the entire genome of individual samples using next-generation sequencing (NGS) and/or third-generation sequencing (TGS), typically at ~5× to 15× coverage [83], without the genome sampling step typical to RRS methods. This approach requires a reference genome sequence for the species under study, or at least for a close relative. The use of a reference transcriptome is also a solution. The possibilities of application of WGRS to many non-model species, especially orphan crops, lacking a reference genome, are limited by this prerequisite. Performing WGRS in such situations requires creating a genome assembly de novo [55]. WGRS provides a very detailed and in-depth insight into the genetic diversity of the analysed germplasm set, since by aligning the sequencing reads of the investigated individuals to the reference genome sequence, all types of genetic variations can be identified, including SNPs, structural variations (SVs), copy number variations (CNVs), presence–absence variations (PAV) and insertion-deletions (InDels). Typically, millions of variations across the genome are identified and scored [55,84]. The ability of variant detection can be limited by the incomplete and inaccurate genome assembly used in data analysis as well as the lack of complete annotation. Although WGRS is highly informative, it is relatively expensive due to the extensive sequencing and data analysis involved [55]. Based on the 2025 fee list from a European core facility CNAG (Centro Nacional de Análisis Genómico, (Barcelona, Spain)) WGRS of a 1 GB genome at 10× coverage typically costs EUR 140–EUR 220 net per sample (library preparation and sequencing).

Typical WGRS bioinformatics analysis starts with checking the quality and trimming reads using tools like FastQC (https://www.bioinformatics.babraham.ac.uk/projects/fastqc/, access date: 2 December 2025) [85] and Trimmomatic (http://www.usadellab.org/cms/?page=trimmomatic, access date: 2 December 2025) [86] to eliminate adapters and low-quality bases. Cleaned reads are then aligned to a reference genome with mappers such as BWA-MEM (https://deepwiki.com/lh3/bwa/3.1-bwa-mem-algorithm, access date: 2 December 2025) [87] or Bowtie2 [88]. After alignment, processing steps like sorting, indexing, and marking duplicates are carried out using SAMtools (https://www.htslib.org/, access date: 2 December 2025) [89] or Picard Tools (https://broadinstitute.github.io/picard/, accessed 7 November 2025) before variant calling. 

For diploid species, calling variants usually involves GATK HaplotypeCaller [40] or bcftools (https://www.htslib.org/, access date: 2 December 2025) [87]. These tools assume there are two alleles per locus and work well when the sequencing depth is sufficient. These workflows are common for crop species like rice, maize, and soybean [90,91,92], where reference genomes are well established. Variants are filtered and annotated using VCFtools [49] and SnpEff (http://pcingola.github.io/SnpEff/, access date: 2 December 2025) [93] to estimate their potential effects.

When analysing polyploid plants, numerous challenges arise, as mentioned above in the RRS Section 2.1. Preferred variant callers, like FreeBayes [39] and GATK4 [94], allow users to define ploidy level and more accurately model allele dosage. For instance, FreeBayes has been effectively used in tetraploid alfalfa to find dosage-sensitive variants and improve genotype accuracy [95].

Genotype imputation is often used to improve SNP density and accuracy in WGRS analysis. Common imputation software options include Beagle (v5.0) [96], IMPUTE2/IMPUTE5 [43,97], practical haplotype graph (PHG) (v2 [98], AlphaPlantImpute (https://github.com/AlphaGenes/AlphaPlantImpute2, access date: 2 December 2025)) [99], and LinkImpute [100]. These tools were developed particularly in non-model organisms and plant datasets with low resources. Success relies on having a well-matched, high-depth reference panel, enough marker overlap between the reference and target samples, and accurate tuning of software settings for plant population structures. It is important to point out that the selection of tools should match the species’ ploidy and population structure (diploid or polyploid; inbred or outbred).

Researchers should recognise that while the imputation software used for WGRS and GBS may be the same, the parameter settings and processing methods differ. For instance, GBS datasets often experience issues with missing genotypes and uneven marker distributions, making it preferable to use imputation software specifically designed to handle missing data. In contrast, for WGRS, the focus should be on variant calling accuracy, depth of coverage, and haplotype representation. These factors should influence the choice of pipelines for SNP calling. Some service providers offer these processing steps, which can alleviate the need for extensive raw data processing on the user’s part.

WGRS has been successfully applied in crops for high-resolution genome-wide association studies, delivering millions of SNPs, capturing rare variants and structural variation [24,83]. Examples of plant germplasm characterisation studies involving WGRS are listed in Table 3.

### 2.3. SNP Genotyping Arrays

High-density DNA genotyping arrays are a hybridisation-based method. DNA samples hybridise to matching oligonucleotide probes attached to the array surface. The detection of a genotype at a given SNP (AA, BB, or AB) is based on emitted fluorescence. SNP arrays allow for genotyping at a fixed set of selected, previously discovered SNPs. Array designs of various densities are available, such as 6 K, 15 K, 60 K, 90 K, or even 1.4 M [51,120,121,122]; therefore, suitable genotyping density can be chosen to match the specific project objectives. While the DNA genotyping arrays were strictly developed to detect a single type of polymorphisms—SNPs—detection of copy number variation (CNV) and presence–absence variation (PAV), as well as analyses of ploidy and aneuploidy are also possible [122,123,124,125].

Design of a genotyping array is a considerable effort, since very large amounts of sequencing data from multiple accessions of a species are necessary to select informative SNPs, whereas factors such as their genomic distribution, minor variant frequency and location relative to genes and non-coding regions should be taken into account [121,126]. For example, to obtain sufficient data for the development of Citrus Genotyping arrays, first, whole-genome sequencing of 41 diverse accessions of *Citrus* and its close relatives was carried out. Then, predominantly genic SNPs from both the nuclear and chloroplast genomes, which were accurately called in *Citrus* and in related species, were selected for the 1.4 M SNP Axiom HD Citrus Genotyping array. Following validation of the HD array, a subset of SNPs was selected for the creation of a 58 K SNP array for more cost-effective genotyping in studies where this lower density is sufficient [121]. Similarly, in pigeonpea, the array development was preceded by WGRS of 105 accessions. This resulted in the discovery of 2.0 M variants from which 56 K were selected for an Axiom array [120].

For SNP arrays, genotype calling is usually performed using specialised software like GenomeStudiorom Illumina (https://support.illumina.com/array/array_software/genomestudio/downloads.html, access date: 2 December 2025) or Axiom Analysis Suite Software 4.0 from Thermo Fisher (Waltham, MA, USA). After the initial calling, quality control and filtering using tools like PLINK [50] or VCFtools [49] are performed.

The same tools used for imputation in GBS and WGRS datasets are also applicable for SNP array data [41,97,99]. For instance, the Infinium Wheat-Barley 40 K SNP array was specifically created to make downstream imputation easier and demonstrated over 99% genotype concordance and less than 5% missing data [127]. The Plant-ImputeDB project [128] compiled multi-species reference panels for plants and showed that genotype imputation can increase SNP density. In rice, an imputation platform combining both SNP-array and whole-genome resequencing datasets, improved the integration of genetic resources and improved the number of usable SNPs for downstream analysis [129].

To date, various genotyping arrays have been developed, mainly for crops with higher economic importance, such as maize (MaizeSNP50 BeadChip [130]), wheat (Wheat 660K SNP array [131]), rice (Rice3K56 [132]), soybean (SoySNP50K [133]), and pigeonpea (Axiom Cajanus 56K [120]).

However, for many plant species, especially for the orphan crops, the requirement of the very large initial financial and computational investment necessary for array development is too prohibitive, and these genotyping options remain unavailable [51]. An important issue to consider when using genotyping arrays is ascertainment bias, which is caused by the selection of SNPs based on analyses of relatively small and not sufficiently diverse populations. The degree of the ascertainment bias is influenced by the size of the population used to discover and select SNPs for the array development (ascertainment or discovery panel) [134,135]. The ascertainment bias can limit the usefulness of SNP arrays in the detection of rare variants in diverse germplasm and identification of introgressions from distant relatives [51]. Furthermore, it was shown that the use of an array with an ascertainment bias results in distorted outcomes of population diversity and structure analyses [134,136]. Other problems, associated mostly with the earlier SNP arrays (developed at a time when the information on the genome sequence, and especially genomic locations of SNPs in the crop in question was not sufficiently detailed) are uneven genomic distribution and redundancy of the SNPs selected for the array design [124]. To mitigate these issues, newer versions of genotyping arrays were developed for various species based on SNP discovery in diverse and large germplasm sets, taking into account the growing knowledge about the genomic distribution of the SNPS and their performance in germplasm characterisation studies. For example, in wheat the TaNG SNP array was developed to address shortcomings of the earlier wheat SNP genotyping arrays: 35 K Wheat Breeders’s array and the Illumina 90 K array—related to the use of small and not sufficiently diverse discovery panels and lack of precise information of genomic SNP location when those arrays were developed—and, at the same time, to take advantages of the advances in the array technology, allowing for a substantial increase in the number of probes in the array. To reach this purpose, skim sequencing data from a total of 315 elite wheat lines and landraces were generated to identify novel SNPs and a haplotype optimisation approach was used to select SNPs for inclusion in the new array. Validated SNPs from previous arrays were also incorporated into the new array. Improved genomic coverage was achieved based on the markers’ positions in the wheat reference genome (IWGSC RefSeq v1.0). The final version of the TaNG SNP array was demonstrated to provide superior results in GWAS compared to the 35 K Wheat Breeder’s array [124] (more details in Section 3).

Once a genotyping array is developed, it constitutes an attractive tool, since in comparison to GBS and WGRS, the computational requirements for data processing are lower, there is also lower missing data and error rate [121,122]. Furthermore, the use of a fixed genotyping array facilitates comparisons across studies. Examples of plant germplasm characterisation studies involving high-density DNA genotyping arrays are listed in Table 4.

### 2.4. Choosing the Right Genome-Wide Genotyping Platform and Other Considerations

With a plethora of methods currently available, selecting the appropriate approach is far from trivial. Various factors—such as the scientific question, budget, species ploidy, and genome complexity—play a decisive part in the decision-making process (Figure 1). In this context, Table 5 presents the main factors to consider and how suitable each method is. In this section, we will discuss some of these factors.

The availability of genomic resources will vary depending on the researcher’s plant species of interest. Thus, the selection of plant species often dictates the options of genotyping methods at hand.

For instance, genotyping rice—a diploid plant with a small, high-quality, well-annotated genome (~450 Mb) and several genomic resources available—germplasm might be a less complicated task than genotyping rye, which has limited genomic resources [141]. A scientist evaluating the population structure of a rice germplasm bank could select SNP arrays, RRS, or WGRS approach to identify polymorphisms. On the other hand, rye scientists might be constrained to RRS genotyping and a few available SNP arrays (rye5 K and 600 K arrays) [142,143], since the large, repetitive rye genome restricts the use of WGRS in this plant species. Pseudocereals and orphan crops with limited genomic resources might face a similar scenario. For example, RRS genotyping was the most suitable approach to evaluate Barnyard millet (*Echinochloa* spp.)—a popular crop in southern and eastern Asia—germplasm diversity since no reference genome or SNP array was available [144,145]. The polyploid nature of Barnyard millet, similar to other orphan crops such as little millet (*Panicum sumatrense*), has held back the development of reference genomes [146]. The previous examples illustrate that the genome complexity and ploidy of a plant species strongly influence the choice of genotyping method. Polyploidy also occurs in several economically important crops such as wheat, oats, potatoes, cotton, and sugarcane [147]. A major challenge in genotyping polyploid species is to distinguish between true SNPs (allelic variation within individuals) and homeologous SNPs (polymorphism among subgenomes within the same individual) [147]. Also, the presence of paralogous loci—duplicated loci resulting from whole-genome duplication—further complicates SNP calling [148,149]. A researcher can increase the correct variant calling rate in a polyploid species either by bioinformatic or sequencing means. Section 2.1 and Section 2.2 of this review discuss some of the tools utilised in SNP calling for polyploid plants.

Long-read sequencing platforms, such as Pacific Biosciences (PacBio) and Oxford Nanopore Technology (ONT), can address the SNP-calling accuracy issue from two distinct angles: reference genome assembly and polymorphism discovery [48,147]. An incomplete or low-quality genome reference containing gaps or sequencing errors leads to poor read mapping [48]. The average read size obtained from both sequencing platforms (25 Kb and 500 Kb for PacBio and ONT, respectively) facilitates the assembly of repetitive regions, generating highly contiguous genomes [150]. Unlike short-read-based reference genomes, long-read genome assemblies provide a more precise landscape of paralogous loci, increasing the accuracy of variant calling. Furthermore, a haplotype-resolved genome assembly can be obtained by combining long-read sequencing and optical mapping or chromatin conformation capture (Hi-C) techniques [151].

The researcher should keep in mind that a single reference genome does not provide the complete picture of genetic diversity within a plant species, even with an available high-quality chromosome-level genome assembly [48,150,152]. The use of pan-genomes—a catalogue of core and accessory sequences from a single species—can capture the complete and accurate genetic variation comprised within a plant (either diploid or polyploid) germplasm [153,154,155]. From the polymorphism discovery standpoint, long-read sequencing permits the identification of SVs that would otherwise go undetected by short-read genotyping approaches [156].

The computational infrastructure should also be taken into account when selecting the genotyping strategy. For instance, determining the population structure of an orphan crop germplasm through WGRS will demand a higher computation capability than an RRS approach [48].

Irrespective of the platform choice, one of the key aspects of any genotyping project is data sharing in conformance with the community accepted, standardised data type formats, to make the data available for reuse in research by the global scientific community. Adherence to FAIR (Findable, Accessible, Interoperable, Reusable) principles [157] is recommended. As a rule, a deposition of the generated data in a recognised publicly available repository is requested before publication of the study results in a scientific journal. An example list of recommended repositories for different data types can be found at https://journals.plos.org/plosone/s/recommended-repositories (accessed on 14 November 2025). For instance, the NCBI Sequence Read Archive (SRA) repository is used for deposition of raw sequencing data, while information on specific genetic variants in a given sample can be submitted (as a VCF file) to the European Variation Archive (EVA). Multiple other repositories, also cross-disciplinary ones, are available.

## 3. Applications of Genome-Wide Genotyping Data in Germplasm Characterisation

Genotyping has become an essential tool in germplasm characterisation and management of genetic resources [6]. Depending on the density and the availability of other datasets (such as results of phenotypic evaluations), genome-wide genotyping data can constitute a basis for a wide range of applications. Genome-wide, high-density genotyping methods deliver very detailed genotypic profiles (composed of hundreds and often tens of thousands of markers covering each chromosome of the genome). Based on such detailed genotypic profiles an evaluation of the genetic diversity levels and their distribution in a given germplasm set can be performed with unprecedented precision to clarify and understand the genetic relationships among accessions. It is possible to determine the population structure—to detect the presence of genetically distinct groups and indicate accessions belonging to each of these groups. When additional data (such as a least passport data) is at hand, the primary determinants of the population structure can be identified, for instance domestication/improvement status, geographic origin [64,77,81], or specific phenotype (for example row type or grain cover in barley [64]).

Application of high-density genotyping methods plays a tremendous role in addressing the challenge of redundancy and error correction in germplasm collections by providing a sufficient number of datapoints to confidently identify groups of duplicates or spot discrepancies between the genotypic information and passport data. It is estimated that unique accessions constitute around 37% of genebank collections worldwide and the unintentional redundancy remains a problem to be addressed to optimise the use of often limited resources available for maintenance and characterisation of genebank collections [1,7]. Recently, based on the GBS genotyping of the barley collection of the Gatersleben genebank, the level of redundancy was found to be around 33% and higher than previously estimated based on passport data [64], while a study using DArTseq successfully identified groups of potentially redundant cassava accessions in the International Center for Tropical Agriculture (CIAT) genebank [7]. These examples underscore the suitability of affordable high-density genotyping (RRS) in the identification of duplicates in both self-pollinated and highly heterozygous crops.

Depending on the study design (if multiple individuals per accession are analysed), the level of homo/heterogeneity of the accessions can be assessed [6]. The within-accession diversity can have a considerable extent, even exceeding the intra-accession diversity, especially in outcrossing species (such as rye or pearl millet [33,158]), but even in self-pollinators, such as barley [64]. Results of such assessments have important implications for the choice of suitable sample sizes and protocols for the periodical regeneration of accessions during gene bank preservation to maintain their genetic integrity.

The comparative analyses of genetic diversity and population structure outcomes and passport data help identify and correct sample tracking errors, for instance, when accessions labelled as “wild” cluster with improved germplasm and are genetically distant from other wild accessions, as was the case in large scale studies of wheat and barley genebank collections, comprising tens of thousands of accessions [64,81]. High-density genotyping data are also used in genetic purity testing, a process that is aimed at the detection of mislabelling, contamination, or residual heterogeneity in seed lots, breeding lines, and hybrid progeny and helps to ensure that these materials are genetically true to their claimed identity [31,159].

Genome-wide high-density genotyping methods are also crucial in supporting the creation of core collections. Core collections represent a subset of germplasm that captures the maximum genetic diversity with minimal redundancy and are essential for efficient conservation, management, and utilisation of genetic resources [140,160].

Another application of high-resolution genotyping data is a selective sweep identification, i.e., detection of genome regions that were subjected to strong positive selection pressure during domestication or plant improvement [77,161]. It provides insights into the evolutionary history of the species and helps to identify loci targeted by selection, which are usually associated with key traits, such as fruit/seed size or seed dispersal mechanisms [77]. The use of high-density genotyping led to a revision of the earlier assumptions about the influence of domestication on the plant genome. While early studies, relying mostly on biparental population and QTL mapping approach, identified few major effect domestication loci [162,163], later population genetics studies using high-density genome-wide genotyping (RRS, WGRS, SNP arrays) revealed that selection pressure impact numerous regions dispersed across the genome and containing hundreds of candidate genes [81,109,164]. Although, as already mentioned in Section 2.1, RRS approaches miss certain genome regions, which might contain genes targeted by specific selection pressures, many population studies employing RRS methods, (such as [77,81,165,166] successfully identified selective sweep regions and candidate genes). However, an important consideration when planning selective sweep detection, is that, similarly like in many other population genetics endeavours, the outcomes depend strongly not only on the type and number of the markers used but also on the composition and size of the germplasm set and on the sweep/outlier detection method/algorithm, hence the results of independent selection scans carried out in the same species often differ [119,167,168]. For example, in tomato, both WGRS and SNP array-based studies [169,170] on domestication identified two stages in the process: domestication stage and improvement stage, with *S. pimpinellifolium* L. as the wild ancestor and *S. lycopersicum* L. var *cerasiforme* as the intermediate species. Razifarad at al. [119] revised this hypothesis using WGRS and a broad collection of *S. pimpinellifolium* L. and *S. l.* var. *cerasiforme* accessions, and found, among others, that the origin of *S. l.* var. *cerasiforme* predated domestication (more details in Section 4).

When the phenotyping data are available, high-density genotyping data of a diverse germplasm set can be used to perform Genome Wide Association Studies (GWAS) to identify causative genes/polymorphisms for a given relevant phenotype and to identify genetic markers associated with key agronomic traits, facilitating marker-assisted selection and crop improvement [83]. Multiple examples of successful GWAS experiments carried out with the use of high-density genotyping data were published to date and several of them were mentioned in Section 4 of this review. However, sometimes the outcomes of the approach are not satisfactory. In some cases, the lack of success can be explained by the shortcomings of the set of markers used for the genotyping. For example, in a comparative analysis, GWAS involving genotyping data generated with the already mentioned (Section 2.3) 35 K Wheat Breeder’s array failed to identify a significant association for any of the three traits: heading data, response to leaf rust, response to stem rust, while the application of the TaNG array-generated genotyping data resulted in the identification of significant associations for each of the analysed traits. While SNPs for the 35 K Wheat Breeder’s array were identified in exome capture data of 43 wheat accessions of various ploidy and, in consequence, a strong ascertainment bias due to the small size of the discovery panel and preferential location of identified SNP in the genic regions was found in subsequent studies involving the use of this array [124,171], the TaNG array was developed based on WGRS data of 315 diverse accessions and provided a better genome coverage [124]. On the other hand, over the last decades it has become apparent that PAV and SV contribute considerably to the genetic and phenotypic diversity of the plant species [172]. In consequence, the use of a single reference genome during GWAS is frequently associated with ‘missing heritability’—a situation when identified marker-trait associations explain only a fraction of phenotypic variation [173]. A study comparing three single-genome references representing three different maize germplasm pools demonstrated that the choice of a reference genome impacts the GWAS outcomes [174]. The availability of a pangenome is a tremendous advantage in such situations. In tomato, the use of a graph pangenome resulted in a 24% increase in estimated heritability in comparison to the single reference genome [173].

## 4. Examples of High-Resolution Genotyping Data Applications in Selected Plant Groups

### 4.1. Cereals

A very exhaustive and comprehensive characterisation of wheat germplasm was carried out by Sansaloni et al. [81] based on GBS (DArTseq) genotyping. The study involved 80,000 accessions—hexaploid and tetraploid wheats as well as wild relatives from CIMMYT (Centro Internacional de Mejoramiento de Maíz y Trigo—International Maize and Wheat Improvement Center) and ICARDA (International Center for Agricultural Research in the Dry Areas) genebanks. In total, ca. 40 to 50 k polymorphic SNP loci (filtering criteria: missing rate ≤ 0.5, minor allele frequency (MAF) ≥ 0.001), were identified within each germplasm group. Data analysis revealed the presence of distinct clusters within the hexaploid wheats and indicated landraces with diversity that is yet to be explored in modern breeding. In tetraploid wheat, a group of ca. 1000 accessions was identified, which were probably misclassified as tetraploids and are in fact hexaploids. The genotyping data were also used to select accessions for core collections constituting ca. 20% of the original germplasm sets, discovery of selection footprints, and, with inclusion of phenotyping data for 3787 accessions, for identification of loci associated with grain protein content and sodium dodecyl sulphate sedimentation (indicative of gluten quality) via GWAS.

WGRS study of an einkorn (*T. monococcum*) diversity panel comprising 219 accessions resulted in the identification of over 121 M of high-quality SNPs. It revealed the existence of three genetic clusters within wild einkorn, corresponding to races α, β and γ, a fairly high diversity within domesticated einkorn and provided support for the hypothesis that einkorn was domesticated from a population related to β race accessions [108].

A breakthrough study by Milner et al. [64] reports on the genetic characterisation via GBS of the whole barley germplasm collection of the IPK Gatersleben genebank and is an excellent example of genebank genomics and its potential. The genotyping of the collection comprising both wild and domesticated barleys (22,626 accessions in total) identified over 170 k polymorphic SNPs (missing rate < 10%, MAF < 1%) and revealed that population structure in barley is largely influenced by the geographic origin, and also by the winter or spring growth habit and morphological characters related to end-use quality (number of rows, and the presence or absence of grain cover). Furthermore, a higher-than-expected level of genetic redundancy in the collection (arising from the presence of duplicate accessions), the extent of within-accession diversity, and errors in passport data were discovered.

A very diverse set of 478 rye (*Secale cereale* L.) accessions was genotyped using DArTseq [77]. As a result, phylogenetic relationships within the *Secale* genus were revealed based on 12.8 k high-quality SNPs (missing data < 10%, MAF > 0.01, reproducibility > 95%), as well as the presence of genetic clusters within cultivated ryes. The study indicated the potential of landraces as a source of new variation for rye breeding and identified putative loci under selection in cultivated germplasm. WGRS of 116 rye accession of worldwide origin and various domestication status resulted in the unravelling of the domestication history of rye, based on 908.6 k SNPs [116], while in another study [26] involving resequencing of 94 diverse rye accessions, which focused mainly on phosphate transporter genes, 820 SNPs within these gene families members, including 12 putatively deleterious variants were identified (out of the total number of 2.5 M SNPs).

In oat (*Avena sativa* L), iSelect 6 K-beadchip was used to analyse a germplasm set comprising almost 290 accessions (mostly landraces) of worldwide origin. Based on 2213 high-quality SNPs, a presence of a relatively strong population structure reflecting geographic origins was discovered, conversely to earlier studies of oat genetic diversity. GWAS involving phenotypic data, available in the Germplasm Resources Information Network (GRIN), identified nine SNPs significantly associated with grain hull type and lemma colour [139]. An interesting approach was used to investigate diversity and population structure in a global collection of cultivated and wild hexaploid oat accessions. GBS data from 15 published and unpublished studies were combined, curated and reanalysed which resulted in the identification of 19,928 SNPs (missing data < 20%, MAF ≤ 1% and heterozygosity ≥ 5%) segregating in 8816 oat taxa. Numerous distinct subpopulations were identified in the analysed set: four subpopulations of wild species *A. sterilis*, a subpopulation of cultivated *A. byzantine* and 16 subpopulations comprising mostly cultivated *A. sativa* accessions. The study also provided support for the role of large-scale chromosome translocations and inversions in shaping population structure and in local adaptation [72].

The Illumina MaizeSNP50 BeadChip, comprising ca. 56 k SNPs, was used for characterisation of 982 maize inbred lines and 190 accessions of teosinte—the wild progenitor of maize. The study revealed phylogenetic relationships of teosinte species, ca. 400 selective sweeps related to maize domestication and 360 adaptive sweeps, resulting from the cultivation of domesticated maize in regions with different environmental conditions (tropical vs. temperate) [130].

Recently, a new and improved rice SNP genotyping array Rice3K56 was developed to address limitations of several previous rice SNP genotyping arrays, such as insufficient marker density or genome coverage. The design of this array was based on WGRS data of over 3 k rice accessions of worldwide origins and included 56,606 high-quality SNPs. The performance of the new array was tested on 192 rice accessions representing both *indica* and *japonica* types and diverse geographic origins. The array turned out to be suitable for varietal identification even when closely related accessions were studied. Similarly, satisfactory outcomes were achieved when GWAS was performed with the new array—over 100 highly significant SNPs associated with 13 traits were identified [132].

### 4.2. Pseudocereals

In recent years, quinoa inbred lines, diverse panels, and wild relatives have been genotyped through GBS and WGRS. For instance, Mizuno et al., [80] analysed 136 inbred lines—using a GBS approach—and identified 5763 SNPs (missing rate < 0.2, MAF > 0.05) showing that these inbred lines fall into three subpopulations (High—Northern and Southern—and lowlands). In another study, Patiranage et al. [25], genotyped a diverse panel of 303 quinoa accessions and seven wild relatives using the WGRS strategy. The study revealed 2.9 M SNPs and associated 600 SNPs with 17 agronomically important traits.

Amaranth is the pseudocereal with genomic tools available, including dedicated genomic databases such as AmaranthGDB [175] and AmaranthGRD [176]. In *Amaranthus*, WGRS of 108 domesticated accessions and wild relatives identified 1.4 M SNPs and aided in elucidating gene flow patterns between domesticated, locally adapted, and wild relatives [101]. Chauhan et al. [60] analysed the population structure of an *Amaranthus* diverse panel, comprising 192 accessions from different parts of the world, using a GBS approach. The study generated 41,931 SNPs (missing rate < 20%, MAF > 5%) and identified three subpopulations in the accession panel.

### 4.3. Tuberous Plants

A collection of 730 accessions from the US Potato genebank, representing various species, was interrogated using GBS [76]. Sets of segregating SNPs ranging from 4.7 to 7.8 k (missing data < 10%, MAF > 0.02) were used for analyses depending on the germplasm set. Higher heterozygosity levels were found in tetraploid potatoes compared to diploid potatoes. Although an overall low population structure was observed, a distinction between wild and cultivated accessions was apparent. Also, indications of introgressions from *S. bolivense* into the cultivated potatoes were found. To support breeding efforts, core subsets consisting of 329 accessions were identified using two algorithms.

DArTSeq genotyping was performed on a collection of 100 winged yam breeding lines [82], and as a result, almost 7 k segregating SNPs (no missing data, MAF = 0.05) were identified. In parallel, phenotyping evaluation involving 24 traits was carried out. Diversity analyses indicated a considerable variation level in the analysed set and the presence of three distinct groups, which is expected to support the selection of parental lines for maximising heterosis in yam breeding.

The largest collection of cassava (*Manihot esculenta*) germplasm, safeguarded at the International Center for Tropical Agriculture (CIAT) cassava genebank was genotyped using the DArTseq method to optimise the detection of genetic redundancy. The quality of DNA samples and of DNA markers (MAF ≥ 0.001, call rate ≥ 0.8) was taken into consideration while defining genetic distance thresholds to identify subsets of genetically distinct accessions. In the characterised set of 5301 accessions (95% of the whole CIAT collection) ca. 2500 (47%) distinct genotypes were identified, with clusters of putatively redundant accessions counting as many as 87 entries [7].

### 4.4. Legumes

A comprehensive chickpea genomic diversity study was performed based on WGRS of 3366 wild and cultivated accessions. A total of 23.51 M SNPs were identified. One of the many outcomes of the study is the identification of 205 SNPs associated with 11 traits. This was achieved by combining phenotypic data of almost 3000 cultivated accessions with genotypic information from almost 4.0 M SNPs. Based on the genomic locations of these SNPs, 79 genes with potential roles in the determination of seed size and development and 24 superior haplotypes for 20 of these genes were identified. Selective sweep detection was also performed, and resulted, among others, in the identification of 37 genes potentially involved in adaptation of chickpea to different cultivation environments. Notably, the study involved the construction of a chickpea pangenome comprising over 3000 individuals [29].

Genotyping data from 2201 accessions from the cowpea core collection, held at the International Institute of Tropical Agriculture (IITA), generated using the Illumina Cowpea iSelect Consortium Array and comprising almost 50 k segregating SNPs, was used to analyse genetic diversity and population structure. In total, 130 groups of putatively identical accessions (comprising up to 15 accessions) were identified. The presence of two major clusters corresponding to geographic origins (West and East Africa) was discovered, with the West African germplasm exhibiting the highest diversity. A successful confirmatory GWAS of seed coat pigmentation patterning was also performed to demonstrate the utility of the collection and accompanying genotyping data for candidate gene identification [138].

The DArTseq method was applied to characterise an existing *Pisum* core collection established at the Instituto de Agricultura Sostenible (Córdoba, Spain), consisting of 325 pea accessions (landraces, wild species, breeding lines, and commercial varieties), and it yielded a total of 11,511 SNP and 24,279 SilicoDArT (presence-absence) markers (missing data < 20%, MAF > 5% and heterozygosity < 10%). Phylogeny analysis revealed six distinct genetic clusters in the collection and provided support for the classification of Pisum into two species *P. flavum* and *P. sativum*. High admixture levels were observed, which inferred continuous genetic exchange among populations. *P. sativum* subspecies: *jomardii* and *arvense* were found to act as introgression channels of wild alleles into cultivated peas during domestication [74].

Another RRS variant, GBS, was used in faba bean to analyse, among others, a diversity panel consisting of 217 accessions, mostly domesticated, representing various geographic origins, and to perform GWAS. Almost 40 k high-quality SNPs were identified (missing rate < 10%, heterozygosity < 10%), and significant maker-trait associations for each of the three traits analysed were found, confirming the suitability of GBS data for this approach [52].

In soybean, WGRS was performed to provide a comparative characterisation of accessions from Kazakhstan and a global germplasm set (684 accessions in total). Almost 81 k high-quality SNPs (missing data < 10%, MAF > 5%) were identified. Population structure and genetic diversity analyses clearly separated wild accessions from the domesticated ones, revealed proximity of Kazakhstani accessions to cultivars from Europe and North America, and, at the same time, a narrow genetic base of Kazakhstani germplasm, providing important cues for local breeding programs [117].

### 4.5. Oil Plants

A study of genetic diversity in South-Central African oil palm germplasm (478 individuals) was performed using GBS. In total, 7048 high-quality SNPs (missing data < 20%, MAF > 0.01) were identified. Population structure analysis indicated the presence of six populations; among them, the Nigerian subpopulation was found to be the most diverse. Finally, 96 palms (individuals), capturing the majority of present alleles, were selected to form a core collection [73].

To aid GWAS in canola (*Brassica napus* L.), a diversity panel consisting of 433 primarily winter populations originating predominantly from North America and Europe was assembled and genotyped using GBS. In total, 251.5 k SNPs (MAF > 0.05), which mapped to the canola reference genome were identified. A considerable genetic diversity of the population was revealed, as well as the presence of genetic clusters. Phenotypic evaluation confirmed ample variation for the several traits evaluated in the panel, suggesting it is suitable for the detection of marker-trait associations [66].

### 4.6. Vegetables

In carrot, WGRS of 630 diverse accessions (wild carrots, cultivars, landraces, and outgroups) was performed to analyse the influence of domestication and improvement. Population structure analyses based on over 168 k high-quality SNPs indicated the existence of five subpopulations corresponding to improvement status and geographic origin. Levels of diversity were found to coincide with the improvement status, with the highest diversity in wild carrots. Indications of a bottleneck, probably related to domestication and recent expansion, were found in each subpopulation, except for the wild carrots. In total, 18 selective sweeps related to domestication and improvement were found, bearing genes related to, among others, photoperiodism, control of flower development and flowering time [103].

A very comprehensive study of pepper (*Capsicum* ssp.) genome diversity, based on resequencing of a core collection comprising 500 accessions representing various wild and domesticated species, provided insights into genetic diversity structure, differentiation, and domestication in this genus. Analyses involving 29 k high-quality SNPs (missing data rate ≤ 0.3, MAF ≥ 0.05) indicated, among others, that the domestications of five cultivated *Capsicum* species occurred independently. Furthermore, selective pressure toward enlarged fruit size and elongated fruit shape targeted distinct genomic regions in different *Capsicum* species [114].

Similarly, in tomato, the WGRS data of 295 *S. pimpinellifolium* L. (wild), *S. lycopersicum* L. var. *cerasiforme* (semidomesticate), and *S. lycopersicum* L. var. *lycopersicum* (cultivated) were a basis to revise the theory regarding its domestication. The population genomics analyses based on over 18 M of SNPs indicated that *S. lycopersicum* L. var. *cerasiforme* (considered to be the semidomesticated intermediate in the evolution of cultivated tomato) exhibited traits associated with cultivated tomatoes, but these traits were lost during the expansion of *S. lycopersicum* L. var. *cerasiforme* forms towards the North and then regained, before the geographic expansion of the cultivated tomato. The study also highlighted the importance of sufficiently broad sampling and carefulness in defining populations in analyses of domestication history [119].

### 4.7. Ornamentals

Analyses based on WGRS (at an average depth of ca. 6.3×) of 525 *Ginkgo biloba* genomes from 51 populations across the world resulted in the identification of over 160 M of high-quality SNPs and revealed multiple cycles of population expansions and reductions. Four subpopulations of ginkgo were identified, as well as population subdivision corresponding to geographical distribution within the Chinese germplasm. Selective sweeps, likely resulting from adaptation to environmental changes, and multiple candidate genes were also detected [109].

Iranian Foxtail lily (*Eremurus*) germplasm consisting of 96 accessions representing seven *Eremurus* species was analysed using GBS, which resulted in 3002 high-quality SNPs (missing data < 50%, MAF > 0.1 and <0.9). Presence of genetic clusters corresponding mostly with the classification into species was found. However, support for the reclassification of two species into subspecies was also provided. Another very useful outcome of this study is the identification of private alleles that can be used to classify foxtail lily at the species level [70].

In *Phalaenopsis,* over 113.5 k of SNPs resulting from GBS genotyping were used to perform GWAS for floral aesthetic traits on a F1 progeny of the cross *P. aphrodite* × *P. equestris* comprising 116 individuals. Ten SNPs for colour-related traits were successfully identified [75].

Using WGRS data from 220 varieties, a 21 K SNP genotyping array was developed to support ornamental *Camellia* breeding. The array proved to be useful in the identification of closely related cultivars and in GWAS. The GWAS analysis was performed for five leaf traits and yielded results consistent with the outcomes based on WGRS data. Population structure analysis of 69 accessions identified four major subgroups in accordance with the available information on the cultivars [137].

### 4.8. Fruit and Nut Trees

RAD-seq was applied to analyse a collection of 168 cultivated and wild apricot accessions. Assessment of genetic diversity and population structure based on over 418 k high-quality SNPs (missing data < 0.5, MAF > 0.05) identified five subpopulations, gene flow between the subpopulations, and selective sweeps caused by domestication [61].

WGRS of 472 accessions representing 48 *Vitis* species from diverse geographic origins was carried out with the aim of providing support for cultivar improvement by, among others, identification of target genes. In total, over 37.8 M of SNPs and 904.4 k of indels (missing calls < 40%, MAF > 0.005) were identified. The population structure of the collection coincided with the improvement status and geographic origin. Genotypic information was also used to perform pedigree analysis and detect genome regions targeted by selection pressure in wild and domesticated grapevines. In total, over 1.3 k candidate genes were found, with only 18 genes in common for both germplasm groups. The study also resulted in the identification of candidate genes for berry shape, panicle type aromatic compounds, and other traits [110].

A collection of 815 common walnut (*Juglans regia* L) accessions, originating predominantly from China, but also from Iran and Pakistan, was analysed using WGRS resulting in the identification of 16.78 M SNPs (MAF > 0.05). Four genetic clusters were identified within the collection, corresponding to the geographic origin of the samples. The level of genetic diversity varied within subpopulations suggesting a genetic bottleneck in Chinese *J. regia* populations. Additionally, numerous signatures of selections related to adaptation and improvement were discovered. The study also involved GWAS analysis based on phenotypic evaluation of Chinese walnut accessions. This resulted in the identification of genomic loci related to 18 key agronomic traits, such as linoleic acid content [177].

In cultivated strawberry, genotyping data comprising almost 30 k SNPs (missing data < 10%, MAF > 0.05) was used, alongside pedigree data, to support the development of a core collection from a set of 920 genotypes from the current breeding program. The genomic data was compiled from Istraw35 SNP array genotyping (available for 891 genotypes) and resequencing results (for the remaining 29 genotypes). The final core collection consisted of 192 genotypes [140].

### 4.9. Others

WGRS with 10× coverage was applied to analyse genomes of 240 *Gossypium barbadense* accessions. In total, over 3.6 M high-quality SNPs and over 220 k indels (missing data ≤ 10%, MAF ≥ 0.05) were identified. The germplasm set under study was found not to be very structured; however, five subgroups could be distinguished. Following GWAS, candidate genes for fibre strength and lint percentage were identified, which could be altered by genetic engineering to speed up cotton improvement [106].

Population genetics analyses based on ca 12 M SNPs (missing data < 30%, MAF > 0.05) identified in WGRS data of 110 cannabis accessions revealed the presence of four distinct clusters in the analysed germplasm and provided insights into the domestication history of the crop, indicating, among others, a single domestication origin of *C. sativa* in East Asia. Multiple candidate genes targeted by selection during the divergence of hemp and drug types were also identified [111].

In tobacco, WGRS of 437 accessions yielded over 2.2 M high-quality SNPs (missing data < 0.1, MAF > 0.05). The germplasm set could be divided into eight subgroups based on population structure analyses. Extensive gene flow between tobacco taxa was also found. GWAS analysis using phenotype data of 379 accessions regarding plant height resulted in the identification of three candidate genes, demonstrating the usefulness of the genomic data resource in assisting molecular breeding [118].

## 5. Conclusions and Outlook

Genome-wide genotyping has become accessible and affordable to the plant community. Researchers and breeders can use WGRS/RRS/SNP to generate high-density marker data for diversity studies, trait mapping, and genomic selection across a wide range of crops, even at large population sizes. The continuous development of bioinformatics tools permits imputation even in large germplasm collections. The reference genome bias and imputation quality in distant germplasm hinder the accuracy of high-density genotyping. Benchmark methods in crops and cross-species comparisons have shown that errors often appear around heterozygous and poorly mapped sequences [178]. Similarly, low-quality and non-contiguous reference genomes of polyploid and orphan crops contribute to poor read mapping. Requirements regarding computational infrastructure and specialised bioinformatic expertise necessary for data analysis and data storage can be limiting for many potential users.

The extent (in terms of numbers of accessions analysed) and scope of germplasm characterisation studies carried out nowadays with the use genome-wide, high-density genotyping vary very much, depending on the crop (its economic importance, genome size and composition, biology), resources available to the research team, the specific study aim, and other factors—from studies exhaustively characterising whole genebank collections comprising tens of thousands of accessions, to very focused analyses of a relatively small set of unique regional germplasm, addressing a specific research question.

Phenotyping is a limiting factor because manual measurements are still the standard for many traits and for calibrating models and are often difficult to repeat across different laboratories [179]. To address this issue, phenomics has emerged, which refers to the high-throughput phenotyping systems (HTP). HTP techniques combine cutting-edge robotics imaging and aerial or spatial platforms of sensor networks with computational tools to capture plant traits across large populations and time scales. This technology enables researchers to detect phenotypic data; for example, water stress or nutrient deficiency, or disease outbreaks at an early stage [180]. Yet, they require careful calibration, standardised workflows, and spatial genotype and environment interaction (G × E) modelling to transform images into interpretable and biologically relevant traits [181,182]. Phenomics, when integrated with machine learning, enables the detection of high-dimensional patterns in phenomics, genomics, and environmental data. These advanced technologies provide accurate prediction of plant performance, trait classification, and automated image analysis [183]. This combination of technologies accelerates the germplasm evaluation and phenotype–genotype association, which strengthens predictive breeding [184].

Increasingly, various emerging innovative technologies are being incorporated into plant germplasm research. Techniques such as single-cell transcriptomics, spatial transcriptomics, and spatial metabolomics are being applied to analyse various aspects of plant biology [185,186] and help understand the impact of environmental stimuli on the phenotypic plasticity [187]. To fully take advantage of the arising wealth of information and apply it effectively for germplasm improvement, an approach called panomics, which integrates genomics, transcriptomics, metabolomics and phenomics and employs deep learning has been proposed [187].

As presented in the current review, genome-wide high-density genotyping powered by advances in NGS technologies has provided the means to significantly increase the information about genetic variation, allele diversity, and gene pool held within the germplasms of important crops. Such information, coupled with the available phenotype data, has identified several agricultural-trait and stress-tolerant-related genes with potential application in Marker-Assisted Selection (MAS) not only in major staple crops but also in underutilised crops. As more germplasms are thoroughly characterised, we will fully tap into the genetic reservoir of such collections and improve our ability to secure food demands in the ever-changing climate conditions.

## Figures and Tables

**Figure 1 ijms-26-11833-f001:**
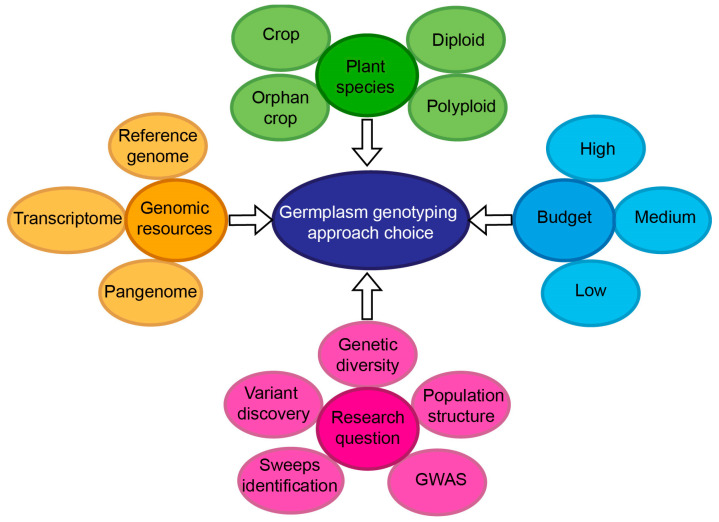
Factors influencing the choice of the genotyping approach.

**Table 1 ijms-26-11833-t001:** Overview of genome-wide high-density genotyping approaches.

Genotyping Approach	Platform	Polymorphism Detection Capacity	Type of Polymorphism Detected	Reference Genome Required	Advantages	Disadvantages
RRS *	GBS DArTseq RAD-seq SLAF-seq	Thousands to hundreds of thousands of SNPs	SNPs PAV(some platforms)	No	Cost-effective. No need for prior SNP information. Works across any genome size and any species. Ideal for non-model and orphan species.	Demands higher computational skills and QC than arrays. Mapping bias. Difficulty distinguishing homologous vs. homoeologous loci in high-ploidy species. Reproducibility is protocol-dependent.
WGRS	Low-coverage-Illumina based Long-read-sequencing based	Millions to hundreds of millions of SNPs	SNPs InDels CNV SVs PAV	Yes	Discovery of various types of polymorphism (SNPs, InDels, CNV, SVs). Survey of polymorphism across the whole genome.	Requires a reference genome sequence (or transcriptome). Demands specialised bioinformatic skills. More expensive compared to RRS and SNP arrays.
SNP arrays	Illumina Infinium (iSelect/BeadChip) Thermofisher Axiom	iSelect HD: 3 k to 90 k iSelect HTS: 90 k to 700 k Up to 2.6 million	SNPs InDels CNV SVs PAV	Yes	Accurate genotyping in polyploid species. Straightforward downstream analysis. High reproducibility and accuracy of genotyping calls. Same SNP calls remain stable across breeding programmes and years	No novel variant discovery. Reduced power to detect marker-trait association Skewed allele frequencies Biased representation of genetic variations

* Abbreviations used in Table 1: CNV—copy number variation, DArTseq—diversity array technology sequencing, GBS—genotyping-by-sequencing, InDel—insertion/deletion, PAV—presence/absence variation, RAD-seq—restriction site-associated DNA sequencing, RRS—reduced representation sequencing, SLAF-seq—specific-locus amplified fragment sequencing, SNP—single nucleotide polymorphism, SV—structural variation, WGRS—whole genome resequencing.

**Table 2 ijms-26-11833-t002:** Examples of RRS application in plant germplasm characterisation studies.

Organism	RRS Variant	No. of Accessions	Institution, Country, Where Germplasm Is Kept	No. of Polymorphic Loci	Analyses	Reference
African mahogany	GBS	115	forest plantations in the Reserva Natural Vale and Viveiro Origem, Brasil	3.3 k	diversity assessment, population structure	[59]
amaranth	GBS	192	National Bureau of Plant Genetic Resources, India	42 k	phylogeny, population structure	[60]
apricot	RAD-seq	168	Luntai National Fruit Germplasm Resources Garden; Yingjisha County Apricot National Forest Germplasm Bank; Xiongyue National Germplasm Resources Garden, China	418 k	population structure, gene flow selection scan	[61]
avocado	GBS	384	Colombian Germplasm Bank; Seedling Rootstocks (SR) (n = 240) of commercial orchards from the northwest Andes; Colombia	4.9 k	diversity assessment, population structure, phylogeny	[62]
banana	DArTseq	856	International Institute of Tropical Agriculture, Nigeria, Tanzania, Uganda; National Agriculture Research Organization, Uganda; Embrapa, Brasil; National Research Centre for Banana, India; International Transit Centre, Belgium	6.1–19.7 k	diversity assessment, population structure	[63]
barley	GBS	22.6 k	Leibniz Institute of Plant Genetics and Crop Plant Research, Germany; National Crop Genebank of China, China; Agroscope, Switzerland	170 k	population structure GWAS redundancy	[64]
blueberry	GBS	195	Philip E. Marucci Center for Blueberry & Cranberry Research and Extension, State University of New Jersey, USA	60.5 k	Population structure, gene flow, section scan	[65]
canola	GBS	433	Kansas State University, USA	251.5 k	population structure	[66]
*Capsicum*	GBS	283	AGROSAVIA La Selva Research Station, Colombia	68.5 k; 30 k	population structure, GWAS	[67]
cassava	DArTseq	5.3 k	International Center for Tropical Agriculture, Colombia	7 k	redundancy	[7]
common bean	GBS	78	International Center for Tropical Agriculture, Colombia	23.3 k	kinship, population structure, GWAS	[68]
*Crotolaria*	GBS	80	Genetic Resources Research Institute of Kenya, Kenya	9.8 k	diversity assessment, phylogeny, population structure	[69]
faba bean	GBS	217	ICARDA genebank, Lebanon	40 k	linkage mapping GWAS	[52]
foxtail lily	GBS	96	wild Eremurus populations in Iran	3 k	phylogeny, population structure	[70]
melon	GBS	755	National Agriculture and Food Research Organization, Japan	39.3 k	diversity assessment, population structure, core subset selection	[71]
oat	GBS	9112	Multiple institutions	19.9 k	population structure, structural rearrangements	[72]
oil palm	GBS	478	Malaysian Palm Oil Board Research Station, Malaysia	7 k	population structure, core subset selection	[73]
peas	DArTseq	325	Instituto de Agricultura Sostenible, Spain	35.8 k	phylogeny, population structure LD scan	[74]
phalaenopsis	GBS	116	National Cheng Kung University, China	113.5 k	GWAS	[75]
potato	GBS	730	US potato genebank, Sturgeon Bay, USA	7.8 k	ploidy estimation, population structure, core subset selection	[76]
rye	DArTseq	478	Several genebanks, universities and breeding companies	12.8 k	phylogeny, population structure, selection scan	[77]
sesame	GBS	501	US Department of Agriculture sesame collection, USDA-ARS Plant Genetic Resources Conservation Unit, USA	24.7 k	phylogeny, population structure, LD scan	[78]
sunflower	RAD seq	135	Active Germplasm Bank of Instituto Nacional de Tecnología Agropecuaria Manfredi, Argentina	11.8 k	diversity assessment, population structure, LD scan	[79]
quinoa	GBS	136	Germplasm Resources Information Network of the US Department of Agriculture, USA	5.7 k	phylogeny, population structure, LD scan	[80]
wheat	DArTseq	80 k	International Maize and Wheat Improvement Center, Mexico; International Center for Agricultural Research in the Dry Areas, Morroco	40 k	population structure, redundancy, core subset selection, selection scan, GWAS,	[81]
yam	DArTseq	100	International Institute of Tropical Agriculture, Nigeria	7 k	population structure	[82]

**Table 3 ijms-26-11833-t003:** Examples of WGRS application in germplasm characterisation studies.

Organism	Coverage (Approx.)	No. of Accessions	Institution, Country, Where Germplasm Is Kept	No. of Polymorphic Loci Detected/(Used)	Analyses	Reference
amaranth	not specified	108	US Department of Agriculture Agricultural Research Service genebank, USA	1.4 M	gene flow, selection scan	[101]
avocado	4.69×	205	Avocado ‘Plus Tree’ Collection; Arangro Plant Nursery; Colombian Germplasm Bank, Colombia	64 M	phylogeny, population structure, racial tracing	[102]
carrot	not specified	630	Germplasm Resources Information Network of the US Department of Agriculture, USA	5.4 M (168 k)	population structure, selection scan, GWAS	[103]
chickpea	12×	3366	International Crops Research Institute for the Semi-Arid Tropics, India; International Center for Agricultural Research in the Dry Areas, Lebanon	23.5 M	GWAS, LD scan, selection scan	[29]
coffee	not specified	90	Choche germplasm bank of the Ethiopian Biodiversity Institute, Etiopia	11 M	phylogeny	[104]
common bean	not specified	144	International Centre for Tropical Agriculture, Colombia; Leibniz Institute of Plant Genetics and Crop Plant Research, Germany; JungleSeeds, Betchworth, UK; Beans and Beans, Horningsham, UK	20.2 M	population structure, phylogeny, GWAS	[105]
cotton	10.85×	240	Zhejiang University, China	3.8 M	phylogeny, population structure, GWAS	[106]
durian		114	cultivations sites in Hainan and Yunnan, China	39 M	diversity assessment, population structure, LD scan, selection scan, core subset selection	[107]
einkorn	not specified	219	Wheat Genetics Resource Center, USA	121 M	phylogeny, population structure	[108]
ginkgo	6.3×	525	Trees growing in multiple locations in China, Japan, Korea USA and Europe	160 M	phylogeny, population structure, selection scan	[109]
grapevine	15.5×	472	Chinese Academy of Sciences; Chinese Academy of Agricultural Sciences, China; Karlsruhe Institute of Technology, Germany	38.7 M	phylogeny, population structure, LD scan, pedigree analysis, selection scan, GWAS	[110]
hemp	10×	110	Vavilov Institute of Plant Genetic Resources, Russia; various companies	12 M	phylogeny, population structure, selection scan	[111]
lettuce	18.8×	445	Centre for Genetic Resources, the Netherlands	208 M	phylogeny, population structure, selection scan, GWAS	[112]
napier grass	15–20×	450	International Livestock Research Institute, Ethiopia; Embrapa, Brasil; US Department of Agriculture, USA; Kenya Agricultural and Livestock Research Organization, Kenya; Lanzhou University, China	170 M (1 M)	diversity assessment, GWAS	[113]
pepper	14.7×	500	U.S. National Plant Germplasm System, USA; Hunan Academy of Agricultural Science, China	1005 M (29 k)	phylogeny, population structure, selection scan	[114]
*Populus cathayana*	32.3×	438	Chinese Academy of Forestry, China	12.3 M	population structure, selection scan, GEA analysis	[115]
rye	10×	116	Germplasm Resources Information Network, USA; Institute of Crop Science, Chinese Academy of Agricultural Sciences, and other collections	908.6 k	phylogeny, population structure, selection scan	[116]
rye	not specified	94	Several genebanks, universities and breeding companies	2.5 M	gene variants	[26]
soybean	not specified	684	Institute of Plant Biology and Biotechnology, Kazakhstan; Guangzhou University, China	8 M (81 k)	phylogeny, population structure	[117]
tobacco	13×	437	Yunnan Academy of Tobacco Agricultural Sciences, China	2.2 M	phylogeny, population structure, gene flow, GWAS	[118]
tomato	not specified	295	Polytechnic University of Valencia, Spain	28 M (18 M; 8.8 M; 162 k)	phylogeny, population structure, selection scan, LD scan, GWAS	[119]
quinoa	7.8×	303	Leibniz Institute of Plant Genetics and Crop Plant Research, Germany; U.S. National Plant Germplasm System, USA	2.9 M	phylogeny, population structure, LD scan, GWAS	[25]

**Table 4 ijms-26-11833-t004:** Examples of high-density SNP-genotyping array applications in plant germplasm characterisation studies.

Organism	Array Name	No. of Accessions	Institution, Country, Where Germplasm Is Kept	No. of Polymorphic Loci	Analyses	Reference
camellia	Camelia21K	69	Camellia Germplasm 479 Resource Conservation Center of the Research Institute of Subtropical Forestry, China	19.3 k	phylogeny, population structure, GWAS	[137]
citrus	1.4 M SNP Axiom^®^ HD Citrus genotyping array	196	Citrus Variety Collection, USA	729 k	population structure	[121]
citrus	58 K Axiom^®^ Citrus genotyping array	871	Citrus Variety Collection, USA	43 k	phylogeny, population structure	[121]
cowpea	Illumina Cowpea iSelect Consortium Array	2201	International Institute of Tropical Agriculture, Nigeria	48 k	population structure, LD scan, GWAS	[138]
maize, teosinte	Illumina MaizeSNP50 BeadChip	1172	maize breeding programs of the International Maize and Wheat Improvement Center (Mexico), China, USA, Thailand, and Peru	42.2 k	phylogeny, population structure, selection scan, GWAS	[130]
oat	iSelect 6 K-beadchip	288	USDA National Small Grain Collection, USA	2213	population structure, LD scan, GWAS	[139]
pigeonpea	Axiom Cajanus SNP array	103	International Crops Research Institute for the Semi-Arid Tropics, India	51.2 k	phylogeny, population structure	[120]
rice	Rice3K56	192	Anhui Agricultural University, China	not specified	phylogeny, varietal identification, GWAS	[132]
soybean	SoySNP50K	286	United States Department of Agriculture, USA	47 k	selection scan	[133]
strawberry	Istraw35	891	The strawberry breeding program at Fresh Forward B.V., Huissen, The Netherlands	30 k	core subset selection	[140]
wheat	TaNGv1.1	908	Germplasm Resources Unit at the John Innes Centre, UK; USDA Germplasm Resource Information Network, USA; Nations BioResource Project-Wheat genebank, Japan	42.5 k	linkage mapping, CNV analysis, GWAS	[124]

**Table 5 ijms-26-11833-t005:** Decision table for choosing an appropriate genotyping method.

Factors	Methods			Comments
	RRS	WGRS	SNP arrays	
Main research objectives				
1. Diversity/population structure	++++	++++	++ *	* May not work for genetically diverse populations.
2. Novel variant discovery	++ *	++++	+ **	* RRS enables partial variant discovery. ** SNP arrays enable none.
3. GWAS/Genome prediction	+++	++ *	++++ **	* Too expensive because a large number of samples are needed for GWAS. ** The best choice due to high reproducibility and low missing data.
4. Selection scans	++ *	++++	+ **	* Weak for haplotype-based scans. ** Ascertainment bias.
Available resources				
1. Reference genome	++++	++++	++++	
2. SNP arrays	++ *	+++ **	++++	* No reason to choose it over arrays unless you want more discovery. ** Choose it only when full genome resolution is required.
3. None	++++	++ *	-	* A de novo genome assembly is required
What is the available budget?				
1. Low	++++	-	++++ *	* If SNP arrays are available
2. Medium	++++	++ *	++++ **	* Lower coverage WGRS. ** Cost depends on marker density and whether the array is commercial or custom.
3. High	++++	++++	++++	When budget is not a concern, the choice of method depends primarily on the study purpose and sample size.
What is the species’ ploidy?				
1. Diploid	++++	++++	++++	Diploid species tolerate all methods well
2. Polyploid	+++ *	++++	++ **	* RRS is only reliable with fully aware polyploidy pipelines. ** SNP arrays are reliable only if designed for that ploidy level.
Cross-study comparability	++ *	+++ **	++++ ***	* Missing data causes low overlap between SNPs in different studies. ** Only if using the same reference and pipeline. *** Data is consistent across labs, years, and experiments.

++++ = Best option with the highest performance; +++ = Very good option with minor limitations; ++ = Good/acceptable option but not optimal; + = Usable but least preferred; - = Not recommended/poor. *,**,*** refer to the information in the 2nd, 3rd and 4th column.

## Data Availability

No new data were created or analysed in this study.

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
