# Peer review of "Genome-Wide, High-Density Genotyping Approaches for Plant Germplasm Characterisation (Methods and Applications)"

_ijms, 2025, doi:10.3390/ijms262411833_

Round 1
Reviewer 1 Report
Comments and Suggestions for Authors
The manuscript entitled 'Genome-Wide, High-Density Genotyping Approaches for Plant Germplasm Characterisation (Current Status: Methods and Applications)' provides a comprehensive review of recent advances in genome-wide genotyping and its applications in crop improvement.
The authors highlight how high-density genotyping, facilitated by next-generation sequencing technologies and imputation strategies, has become cost-effective for both major and underutilized crops. The review also emphasizes the continuing challenges, such as reference bias, imputation quality in distant germplasm, and limitations in phenotyping. As a future perspective, authors propose the integration of genomic analysis with high-throughput phenotyping. Overall, the manuscript offers a valuable synthesis of current methods and their potential to accelerate breeding programs. Moreover, the authors prove that deep genetic diversity characteristics of different plant species resources can be beneficial for agricultural improvement.
I like the thorough presentation of current methods and their practical applications, which makes the review a valuable contribution to the field. I have only minor comments on the manuscript, which are mainly editorial. The scientific content is proper, and the work is well-structured and clearly presented, requiring no substantive revisions. I would suggest, however, revising the manuscript title to the following: “Genome-Wide, High-Density Genotyping Approaches for Plant Germplasm Characterization: Methods and Applications”, as it sounds more natural.
The Abstract and Introduction are very well written and clear. Chapter three is also well written, offering a varied structure and avoiding the feeling of repetitive patterns. However, I have the impression that the summary, particularly its first two paragraphs, may have been written in a slightly different style compared to the abstract, introduction, or the first chapter. The authors may wish to consider rephrasing this section for consistency. I really like this part of the sentence: …accessions constitute a living library of natural genetic variation…
I have suggested the inclusion of two additional references related to oats, as this is my area of expertise. In particular, one of the proposed studies, published at the end of October this year, is quite relevant. However, I leave the decision to the authors, as I acknowledge that I may not be fully objective on this topic.
Further comments can be found in the attached PDF file.
I believe that the manuscript is suitable for publication after minor revisions.

Author Response
Dear Reviewer,
Thank you very much for your thorough and insightful review of our manuscript.
Below are the point-by-point responses to the specific comments:
“I would suggest, however, revising the manuscript title to the following: “Genome-Wide, High-Density Genotyping Approaches for Plant Germplasm Characterization: Methods and Applications”, as it sounds more natural.”
We have revised the title as suggested
‘However, I have the impression that the summary, particularly its first two paragraphs, may have been written in a slightly different style compared to the abstract, introduction, or the first chapter. The authors may wish to consider rephrasing this section for consistency. “
Thank you for pointing this problem out to us. We have rewritten this part manuscript to make it easier to read and more understandable for broader audience.
“I have suggested the inclusion of two additional references related to oats, as this is my area of expertise. In particular, one of the proposed studies, published at the end of October this year, is quite relevant. However, I leave the decision to the authors, as I acknowledge that I may not be fully objective on this topic”
Thank you for drawing our attention to these research articles. We have chosen to include in our manuscript the study by Bekele et al (2025) , due to its broader scope and the very interesting approach of combining multiple smaller data sets to be able to get a deeper insight and a stronger support for the conclusions.
“Further comments can be found in the attached PDF file.”
We have corrected all indicated errors. We also fixed problems with references by converting the references to the recommended IJMS style.
With kind regards
On behalf of the Authors
Hanna Bolibok-Brągoszewska

Reviewer 2 Report
Comments and Suggestions for Authors
Overall Comments:
This manuscript provides a comprehensive and well-organized overview of current genome-wide, high-density genotyping approaches for plant germplasm characterization. The review is well-structured, timely, and covers the major platforms (RRS, WGRS, SNP arrays) with extensive, up-to-date examples across a wide range of crop species. The content is highly relevant for the field of genetic resources and plant breeding. The manuscript is generally suitable for publication after minor revisions, primarily to address the points below which would further strengthen its depth and perspective.
Specific Comments for Revision:
- Lack of discussion on emerging technologies, such as pangenome-based genotyping, single-cell genomics, spatial transcriptomics, and long-read sequencing (e.g., PacBio HiFi, Oxford Nanopore), which are increasingly applied in germplasm characterization and structural variation detection.
- Limited comparison of bioinformatic pipelines for different genotyping platforms. A brief overview of commonly used software for variant calling, imputation, and quality control would enhance the practical utility of the review.
- Inconsistent depth in case studies: While numerous examples are provided, some lack critical details such as sequencing coverage, SNP filtering criteria, or population size, which are essential for reproducibility and comparative analysis.
- Underemphasis on polyploid and orphan crops: Although mentioned, the specific challenges and solutions for genotyping in polyploid species and under-researched crops deserve a more dedicated discussion.
- Missing forward-looking perspective on the integration of multi-omics data (e.g., genomics with phenomics, metabolomics) and machine learning approaches for predictive breeding and germplasm utilization.
- Recommendation to include a section on data standardization and sharing, such as the use of common ontologies (e.g., MIAPPE, FAIR principles), to facilitate global collaboration and data reuse in germplasm research.
Author Response
Dear Reviewer,
Thank you very much for your insightful review of our manuscript. Below are the point-by-point responses to your comments:
- Lack of discussion on emerging technologies, such as pangenome-based genotyping, single-cell genomics, spatial transcriptomics, and long-read sequencing (e.g., PacBio HiFi, Oxford Nanopore), which are increasingly applied in germplasm characterization and structural variation detection.
To address this comment we have added information about the advantages of using reference genome which was assembled with the long-read sequencing data in a new section of the manuscript – section 2.4 and highlighted the issue of SV detection based on long reads.
We have also underlined the value of a pangenome in germplasm research in section 2.4 and also we have included an example of impact of the use of an pangenome on GWAS outcome in section 3 (Zhou et al. 2022)
We have mentioned emerging technologies such as single cell transcriptomic, special transcriptomics and metabolomics in the section 5
- Limited comparison of bioinformatic pipelines for different genotyping platforms. A brief overview of commonly used software for variant calling, imputation, and quality control would enhance the practical utility of the review.
A description of commonly used bioinformatic pipelines/software was added to each of the manuscript sections describing the genotyping platforms (2.1, 2.2 and 2.3)
- Inconsistent depth in case studies: While numerous examples are provided, some lack critical details such as sequencing coverage, SNP filtering criteria, or population size, which are essential for reproducibility and comparative analysis.
We have revised section 4 to include the missing information, where available
- Underemphasis on polyploid and orphan crops: Although mentioned, the specific challenges and solutions for genotyping in polyploid species and under-researched crops deserve a more dedicated discussion.
Specific challenges associated with genotyping of polyploid genomes and orphan crops were mentioned in the section 2.4. Also bioinformatic tools suitable for analyses in polyploid genomes were specified in sections 2.1-2.3
- Missing forward-looking perspective on the integration of multi-omics data (e.g., genomics with phenomics, metabolomics) and machine learning approaches for predictive breeding and germplasm utilization.
We have added a paragraph in section 5 mentioning the panomic approach, which proposes integration of various types of omics data and deep learning approaches in plant improvement
- Recommendation to include a section on data standardization and sharing, such as the use of common ontologies (e.g., MIAPPE, FAIR principles), to facilitate global collaboration and data reuse in germplasm research.
As requested, we have added a paragraph on data sharing and FAIR principles at the ed of section 2.4
With kind regards, on behalf of the Authors,
Hanna Bolibok-Brągoszewska

Reviewer 3 Report
Comments and Suggestions for Authors
While competent, the manuscript is more of a descriptive catalogue than a critical synthesis. It lacks the depth needed to guide researchers through the complex realities of modern germplasm genomics. The following critical points must be addressed to elevate its impact.
- The manuscript expertly describes each genotyping method and its pros and cons in Table 1 and the accompanying text. However, it would be significantly more impactful if it included a dedicated section or a decision-guiding figure (e.g., a flowchart) to help researchers select the most appropriate method. This guide could be based on key questions such as:
- Is a high-quality reference genome available for the species?
- What is the primary research goal (e.g., diversity analysis, novel variant discovery, GWAS, routine QC)?
- What is the available budget per sample?
- Is the species diploid or polyploid?
- Is cross-study comparability a priority?
- The manuscript tells us that these studies were done but doesn’t adequately explore what we have learned collectively from them. For instance, have WGRS studies consistently overturned long-held beliefs about crop domestication pathways established by older marker systems? Has GBS proven more effective for population structure analysis than for GWAS in outcrossing, heterozygous species?
I would suggest authors to revise Section 4 and/or add a dedicated discussion section that connects the dots. Instead of just listing studies by crop type, consider organizing the discussion around key scientific questions that have been answered or raised by these new technologies. For example:
- "Revisiting Domestication and Diversity with Unprecedented Resolution"
- "The Challenge of Redundancy and Error Correction in Large-Scale Genebank Genomics"
- "Successes and Failures of GWAS in Wildly Diverse Germplasm"
- The point "Reproducibility is protocol-dependent" is a critical understatement. The choice of restriction enzymes, size selection, and library prep protocols can lead to entirely different subsets of the genome being sequenced, making cross-study comparison nearly impossible without re-genotyping everything on one platform. This issue of data incompatibility is a major impediment in the field and deserves more emphasis.
- The need for a "reference genome" is presented as a simple prerequisite. The reality is that the quality of the reference is paramount. A fragmented, poorly annotated, or non-representative reference (e.g., based on a single elite cultivar) can lead to massive mapping bias, incorrect variant calls, and the complete loss of data from novel structural variants or PAVs present in diverse germplasm. The challenges of applying WGRS in polyploid species are also significantly downplayed.
- The review mentions "ascertainment bias" but does not fully unpack its consequences. This bias can render an array virtually useless for studying wild relatives or exotic landraces, as the SNPs were chosen from a narrow panel of elite lines and will be largely monomorphic in distant material. The enormous upfront cost and effort to develop an array, which creates a high barrier to entry for non-major crops, is also not sufficiently highlighted as a major constraint.
- For commnet 4, 5, and 6, I suggest for each method, dedicate a paragraph to a more "warts and all" discussion of these practical pitfalls. Use specific examples where possible (e.g., a study that failed due to reference genome bias or a comparison of two different GBS protocols on the same material).
- The review focuses on data generation and almost completely neglects the immense downstream bottlenecks. I suggest to add a section on the "post-sequencing challenge," covering the critical needs for computational infrastructure, data storage, standardized pipelines, and specialized bioinformatic expertise—the true rate-limiting steps for most genebanks.
- The conclusion that "phenotyping is a bottleneck" is a generic statement that lacks analytical depth. I suggest to move beyond this simple point. Critically discuss the complexity of high-quality phenotyping, the analytical challenges of integrating HTP data, and the crucial impact of G×E (Genotype × Environment) interactions, which are often the primary hurdle in translating genomic data into practical outcomes.
Author Response
Dear Reviewer,
Thank you very much for your insightful and in-depth review of our manuscript. We have revised our manuscript to address the issues raised in you review. Below are the point-by-point responses to your comments:
- The manuscript expertly describes each genotyping method and its pros and cons in Table 1 and the accompanying text. However, it would be significantly more impactful if it included a dedicated section or a decision-guiding figure (e.g., a flowchart) to help researchers select the most appropriate method. This guide could be based on key questions such as:
- Is a high-quality reference genome available for the species?
- What is the primary research goal (e.g., diversity analysis, novel variant discovery, GWAS, routine QC)?
- What is the available budget per sample?
- Is the species diploid or polyploid?
- Is cross-study comparability a priority?
We have added a new section (2.4 ) to the manuscript, where we listed the factors influencing the choice of the genotyping platform and discussed several of them in more detail.
- The manuscript tells us that these studies were done but doesn’t adequately explore what we have learned collectively from them. For instance, have WGRS studies consistently overturned long-held beliefs about crop domestication pathways established by older marker systems? Has GBS proven more effective for population structure analysis than for GWAS in outcrossing, heterozygous species? I would suggest authors to revise Section 4 and/or add a dedicated discussion section that connects the dots. Instead of just listing studies by crop type, consider organizing the discussion around key scientific questions that have been answered or raised by these new technologies. For example:
- "Revisiting Domestication and Diversity with Unprecedented Resolution"
- "The Challenge of Redundancy and Error Correction in Large-Scale Genebank Genomics"
- "Successes and Failures of GWAS in Wildly Diverse Germplasm"
We have decided to keep section 4 mostly unchanged. This section was meat to be an overview of germplasm characterisation studies being carried out nowadays and to show that the extent and scope of these studies vary very much, depending on the crop (its economic importance, genome size and composition, biology), resources available to the research team, the specific study aim, etc.
To address the issues raised by you in point 2 of the review we have expanded section 3. We tried to put more emphasis on the issues of unprecedented resolution in diversity studies, the problem of redundancy and genetic integrity, revising the finding regarding domestication. We have highlighted a specific example of revising the earlier findings regarding the domestication of tomato (Razifarad at al 2020) and described in more detail a situation when GWAS failed due to shortcoming on an array, but was successful with a newer array design (Burridge et al 2024)
- The point "Reproducibility is protocol-dependent" is a critical understatement. The choice of restriction enzymes, size selection, and library prep protocols can lead to entirely different subsets of the genome being sequenced, making cross-study comparison nearly impossible without re-genotyping everything on one platform. This issue of data incompatibility is a major impediment in the field and deserves more emphasis.
We have added more information about the influence of the experimental design on the outcomes of GBS in the section characterising RRS approaches (2.1) (Zamalutdinov et al 2025)
- The need for a "reference genome" is presented as a simple prerequisite. The reality is that the quality of the reference is paramount. A fragmented, poorly annotated, or non-representative reference (e.g., based on a single elite cultivar) can lead to massive mapping bias, incorrect variant calls, and the complete loss of data from novel structural variants or PAVs present in diverse germplasm. The challenges of applying WGRS in polyploid species are also significantly downplayed.
He have added information about the influence of reference genome quality, the advantage of assemblies from long reads in identification of SV, and challenges related to genotyping of polypoids to the new section 2.4. Specific information on software suitable for analysis of data from polyploids was added to sections 2.1 and 2.2
- The review mentions "ascertainment bias" but does not fully unpack its consequences. This bias can render an array virtually useless for studying wild relatives or exotic landraces, as the SNPs were chosen from a narrow panel of elite lines and will be largely monomorphic in distant material. The enormous upfront cost and effort to develop an array, which creates a high barrier to entry for non-major crops, is also not sufficiently highlighted as a major constraint.
We have expanded the paragraphs describing the problems of ascertainment bias (including its influence of the outcomes of population diversity and structure assessments) and initial investment need to develop a SNP panel in the section 2.4. - For comment 4, 5, and 6, I suggest for each method, dedicate a paragraph to a more "warts and all" discussion of these practical pitfalls. Use specific examples where possible (e.g., a study that failed due to reference genome bias or a comparison of two different GBS protocols on the same material).
We specified above (in responses to points 3,4,5) how we addressed this issues while revising the manuscript
- The review focuses on data generation and almost completely neglects the immense downstream bottlenecks. I suggest to add a section on the "post-sequencing challenge," covering the critical needs for computational infrastructure, data storage, standardized pipelines, and specialized bioinformatic expertise—the true rate-limiting steps for most genebanks.
To address the problem of bioinformatic analysis better we have included description of analysis steps, pipelines and tools to each of the sections 2.1, 2.2, and 2.3. We have also added brief mentions regarding computational infrastructure constraints to sections 2.4 and 5.
- The conclusion that "phenotyping is a bottleneck" is a generic statement that lacks analytical depth. I suggest to move beyond this simple point. Critically discuss the complexity of high-quality phenotyping, the analytical challenges of integrating HTP data, and the crucial impact of G×E (Genotype × Environment) interactions, which are often the primary hurdle in translating genomic data into practical outcomes.
We have rewritten this paragraph and added more specific information regarding the constraints of phenotyping we have also included a sentence about panomics concept, proposing integration of different types of omics data in germplasm improvement
With kind regards, on behalf of the Authors,
Hanna Bolibok-Brągoszewska

Round 2
Reviewer 3 Report
Comments and Suggestions for Authors
It is very difficult to review the revised manuscript in its current format. The authors have used track changes in a way that often deletes whole paragraphs and then re-inserts almost identical text with re-formatted references and only minor wording changes. As reviewing is a time-consuming task, this makes it extremely challenging to read the manuscript efficiently and to clearly identify what substantive changes have actually been made.
Author Response
Dear Reviewer,
We are sorry for the shortcomings of the tracking of changes. The removed text is in blue and stroked. The added text is in red and underlined. If other colours or formatting would be beneficial, please let us know.
We have prepared an updated version of the manuscript with tracked changes: We have improved the tracking of the problematic paragraph (the paragraph, which position has been moved and appeared both as removed and added text) about GWAS in section 3, and also revised the tracking of changes of the first two paragraphs in Section 5 - to simplify reading of the text we have placed the original text at the start of the section 5 and the new version of the opening paragraph directly underneath.
The location of major changes (mostly new text) in the new version of the manuscript with tracked changes file is as follows:
Section 2.1
lines 116 – 150 (description of data processing)
lines 159-164 (comparison of protocols and pipelines)
lines 171-176 (short reads can be used in detection of SV if reference suitable)
Section 2.2
Lines 204-232 (description of data processing)
Section 2.3
Lines 260-269 (description of data processing)
Lines278-300 ( ascertainment bias)
Section 2.4 (new section, all new text)
Lines 309-267
Section 3
Lines 382 – 402 (high density genotyping in addressing the issue of redundancy and genetic integrity of accessions)
Lines 419-438 (new findings regarding domestication)
Lines 443 – 463 (high density genotyping in GWAS)
Section 5
Lines 727 -737 (new version of the previous opening paragraphs; the previous version in located in lines 715-726)
Lines 738-744 – new text
Lines 747-759 – revised text, with addition al information about phenotyping constraints, phenomics, etc.
Lines 762 -768 – emerging technologies in germplasm research and the concept of panomics
We hope this information will facilitate the evaluation of the revised manuscript
With kind regards, on behalf of the Authors,
Hanna Bolibok-Brągoszewska

Round 3
Reviewer 3 Report
Comments and Suggestions for Authors
Authors have made significant changes, although they left a few points unanswered, such as the addition of a flowchart diagram or any other form of visual presentation to summarize their review, along with a few minor issues. I would particularly like to see a clear flowchart diagram while reading the manuscript, as it would provide an overview instead of having to go through the entire wordy text to pick up key points.
Moreover, there are a few minor comments below:
- The faba bean GBS study appears twice with different numbering (e.g., [51] and [161]) but essentially refers to the same authors, title, and DOI.
- Several entries contain placeholders like “???” (e.g., Blanca et al., 2015) or missing journal names. These must be corrected.
- Poplin et al. (2017) is cited as a bioRxiv preprint without noting its later formal publication. Please update this citation to the final published version if available.
Author Response
Dear Reviewer,
Thank you very much for your feedback on our revised manuscript. Below are the point-by-point responses to your comments:
"Authors have made significant changes, although they left a few points unanswered, such as the addition of a flowchart diagram or any other form of visual presentation to summarize their review, along with a few minor issues. I would particularly like to see a clear flowchart diagram while reading the manuscript, as it would provide an overview instead of having to go through the entire wordy text to pick up key points.”
To address this comment we have added the Figure 1 to the manuscript, which shows major factors influencing the choice of the genotyping approach.
We have also included the Table 5 listing the factors and indicating suitability of each method, with additional comments. We have decided in favour of this method of the visual presentation of the problem after several unsuccessful attempts to a flowchart. Since several factors have to be considered in parallel while making a choice of a genotyping platform, the flowchart gets very complicated. On the other hand we wanted to avoid an oversimplification.
„Moreover, there are a few minor comments below:
- The faba bean GBS study appears twice with different numbering (e.g., [51] and [161]) but essentially refers to the same authors, title, and DOI.”
We have corrected this issue. We have also noticed a few references in the text which were not included in the text. This is corrected now
- „Several entries contain placeholders like “???” (e.g., Blanca et al., 2015) or missing journal names. These must be corrected.”
Thank you very much for pointing this out. We have added the missing information.
- „Poplin et al. (2017) is cited as a bioRxiv preprint without noting its later formal publication. Please update this citation to the final published version if available."
We have kept the reference to the bioRxiv preprint, since it is recommended by the Creators
https://gatk.broadinstitute.org/hc/en-us/articles/360035530852-How-should-I-cite-GATK-in-my-own-publications. However we noticed a mistake in the title and corrected in. It is now the reference 85 in the revised manuscript.
With kind regards, on behalf of the Authors,
Hanna Bolibok-Brągoszewska
